# Measuring Local and Shuffled Privacy of Gradient Randomized Response

## Abstract

Local differential privacy (LDP) provides a strong privacy guarantee in a distributed setting such as federated learning (FL). Even if deployed LDP mechanisms honestly provide such privacy guarantees by randomizing gradients, how can we confirm and measure it? To answer the above question, we introduce an empirical privacy test in FL clients by measuring the lower bounds of LDP. The results of this measurement give the client the empirical $\varepsilon$ and probability that the two gradients can be distinguished. We then instantiate five adversaries in FL under LDP to measure empirical LDP at various attack surfaces, including a worst-case attack that reaches the theoretical upper bound of LDP. The empirical privacy test with the adversary instantiations enables FL clients to understand LDP more intuitively and verify that mechanisms claiming $\varepsilon$-LDP actually provide equivalent privacy protection. We also demonstrate numerical observations of the measured privacy in these adversarial settings, and the randomization algorithm LDP-SGD is vulnerable to gradient manipulation and a well-pre-trained model. We further discuss employing a shuffler to measure empirical privacy in a collaborative way and also measuring the privacy of the shuffled model. Our observation suggests that the theoretical $\varepsilon$ in the shuffle model has room for improvement.

## 1 Introduction

Local differential privacy (LDP) (Evfimievski et al., 2003; Kasiviswanathan et al., 2011) provides a strong privacy guarantee in a distributed setting such as federated learning (FL). The randomized mechanisms that ensure LDP make it difficult to distinguish randomized responses crafted from any input (e.g., gradient) to the extent that it is quantified by privacy parameter $\varepsilon_0$. Furthermore, the shuffle model (Erlingsson et al., 2019; Cheu et al., 2019), which anonymizes identities of the randomized responses via secure shuffler and then amplifies the LDP, has gained significant interest.

Even if deployed LDP mechanisms honestly provide such privacy guarantees by randomizing gradients, how can we confirm and measure it? According to a previous survey (Xiong et al., 2020), some users who did not allow information sharing claimed two reasons: they did not trust the DP techniques and did not trust the app or tech company. We believe that to encourage more users to contribute data, it is necessary to clearly explain LDP mechanisms and confirm that the randomized mechanism is credible.

Auditing differential privacy of machine learning models has been studied mainly for DP-SGD (Abadi et al., 2016) that employs the Gaussian noise under central DP constraint (Jagielski et al., 2020; Nasr et al., 2021; Maddock et al., 2022; Andrew et al., 2023). In the distributed settings, *gradient randomized response* that randomly samples a vector based on the original gradient vector, such as LDP-SGD (Erlingsson et al., 2020) and PrivUnit (Bhowmick et al., 2018) have been widely utilized in local and shuffle DP models. The gradient randomized response differs entirely from the Gaussian mechanism; therefore, we need another discussion. How can we measure the empirical privacy of the gradient randomized responses in local and shuffle DP settings?

To answer the above question, we introduce an empirical privacy test in FL clients by measuring the lower bounds of LDP. Since (local) differential privacy is designed to quantify the privacy level in the worst-case scenario, we first analyze the worst-case of the gradient randomized responses, especially in LDP-SGD. Hence, we actualize a way to measure the empirical differential privacy

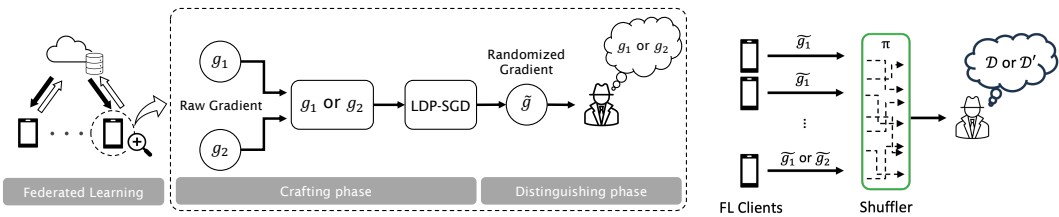

(a) Client-side privacy measurement test in FL.     (b) Privacy test in the shuffle model

Figure 1: Overview of the empirical privacy measurement test for (a) each randomized response and (b) a shuffled batch. Our privacy measurement test consists of the **crafting phase**, which generates (malicious) inputs, and the **distinguishing phase**, which infers the input. Play the game of distinguishing inputs from outputs enough times, and we can get an empirical privacy level $\varepsilon_0$. The game with collaborating $n$ clients enables us to measure the empirical amplified $\varepsilon$ on the shuffler (b).

Table 1: Our measurement test is unique in that it deals with FL under LDP.

| Techniques | Setting | DP model | Mechanism |
|---|---|---|---|
| Jagielski et al. (2020) Nasr et al. (2021) Lu et al. (2022) Bernau et al. (2021) Zanella-Béguelin et al. (2022) | Centralized | CDP | DP-SGD |
| Maddock et al. (2022) Andrew et al. (2023) | FL | CDP | DP-Fed. SGD/Avg. |
| **Ours** | **FL** | **LDP & Shuffle** | **LDP-SGD** |

of LDP-SGD. We then instantiate five adversaries in FL under LDP to measure empirical LDP at various attack surfaces, including a worst-case attack that reaches the theoretical upper bound of LDP.

We further propose to apply our measurement test to the shuffle model and show empirical privacy amplification. The privacy amplification via shuffling has been studied mainly in gradient randomized responses like LDP-SGD but not the Gaussian mechanism. Our empirical test for the randomized responses also enables us to measure the empirical privacy level given by the shuffler. This is also our novelty against the prior studies (Jagielski et al., 2020; Nasr et al., 2021; Maddock et al., 2022; Andrew et al., 2023) that employ the Gaussian mechanism (e.g., DP-SGD).

**Contributions.** The contributions of this study are summarized as follows:

1. We establish a client-side empirical privacy measurement test of LDP in FL. We also introduce a way to measure empirical privacy amplification by shuffling.
2. We discover the worst-case attack in FL using LDP-SGD. The worst case is essential to confirm the privacy guarantees empirically.
3. We demonstrate the empirical privacy levels at various attack surfaces and show that LDP-SGD is vulnerable to gradient manipulation and a well-pre-trained model.

Table 1 shows the differences between our proposed method and related studies. Our technique is designed for FL and differentiates itself from previous studies because it deals with local DP and shuffle DP using the gradient randomized responses.

## 1.1 RELATED WORK

Auditing differential privacy of machine learning models has been studied in centralized settings. Most of the works analyzed the privacy level of training models privatized with DP-SGD (Abadi et al., 2016) against membership inference (Yeom et al., 2018) and two poisoning attacks (Gu et al., 2017; Jagielski et al., 2020). Liu et al. (2019) proposed an interpretation of the CDP using a hypothesis test. The lower bound measurement method devised by Jagielski et al. (2020) allowed the observation of how DP-SGD reached its upper bound when exposed to various threats. Nasr et al. (2021) demonstrated empirical privacy with several instantiated adversaries, including the worst-case against DP-SGD. CANIFE (Maddock et al., 2022) crafts data poisoning canaries adaptively to generate model updates orthogonal to updates sent by other clients in each round. Zanella-Béguelin

**Algorithm 1** LDP-SGD; client-side $\mathcal{A}_{\text{client}}$ (Erlingsson et al., 2020)

**Require:** Local privacy parameter: $\varepsilon_0$, current model: $\theta_t \in \mathbb{R}^d$, $\ell_2$-clipping norm: $L$
1: Compute clipped gradient
$$x \leftarrow \nabla\ell(\theta_t; d) \cdot \min\left\{1, \frac{L}{||\nabla\ell(\theta_t; d)||_2}\right\}$$
2: $z \leftarrow \begin{cases} L \cdot \frac{x}{||x||_2} & \text{w.p. } \frac{1}{2} + \frac{||x||_2}{2L} \\ -L \cdot \frac{x}{||x||_2} & \text{otherwise.} \end{cases}$
3: Sample $v \sim_u S^d$, the unit sphere in $d$ dims
$$\hat{z} \leftarrow \begin{cases} \text{sgn}(\langle z, v \rangle) \cdot v & \text{w.p. } \frac{e^{\varepsilon_0}}{1 + e^{\varepsilon_0}} \\ -\text{sgn}(\langle z, v \rangle) \cdot v & \text{otherwise.} \end{cases}$$
4: **return** $\hat{z}$

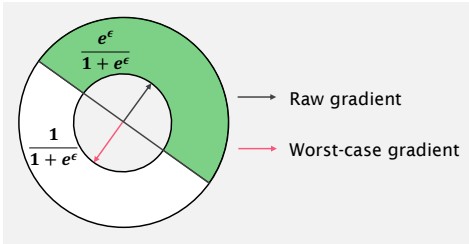

Figure 2: Overview of the gradient randomization by LDP-SGD: the gradient will likely be sampled from the green zone when privacy parameter $\varepsilon_0$ is given a large value.

et al. (2022) and Andrew et al. (2023) proposed efficient hypothesis testing approaches for the privacy auditing. ML Privacy Meter (Murakonda & Shokri, 2020) verifies a given theoretical privacy level through membership inference (Shokri et al., 2017), but does not give empirical privacy level.

The uniqueness of our study is that the target DP mechanism is entirely different from previous studies. Previous studies estimate empirical privacy under the Gaussian mechanism (e.g., DP-SGD and DP-FedAvg). Our work studies how to craft adversaries tailored to the gradient randomized responses that LDP-SGD assumes and also discovers the worst-case scenario in the LDP-SGD. Therefore, the worst-case is entirely different from those of the Gaussian mechanism. Our empirical privacy test also enables us to measure empirical privacy amplification via shuffler. Since most of the shuffling protocols have been discussed under $\varepsilon_0$-LDP with the randomized responses but not $(\varepsilon_0, \delta_0)$-LDP with the Gaussian mechanism, one of the novelty of our proposed privacy test is that it discusses the empirical privacy amplification.

## 2 PRELIMINARIES

This section introduces information that is essential for understanding our proposal. FL assumed in this paper consists of an untrusted model trainer (server) and clients that own sensitive data. First, the client creates the gradient with the parameters distributed by the model trainer. The client then randomizes the gradient under LDP and sends it to the model trainer. The model trainer updates the parameters with the gradient collected from the client.

### 2.1 LOCAL DIFFERENTIAL PRIVACY

Local differential privacy (Evfimievski et al., 2003; Kasiviswanathan et al., 2011) is an extension of differential privacy (Dwork et al., 2006) for privately gathering statistics from distributed data.

**Definition 1** ($\varepsilon_0$-Local Differential Privacy). A randomized mechanism $\mathcal{M} : \mathcal{X} \rightarrow \mathcal{S}$ satisfies $\varepsilon_0$-local differential privacy if and only if, for any pair of inputs $x, x' \in \mathcal{X}$ and for any possible output $S \in \mathcal{S}$, it holds that
$$\Pr[\mathcal{M}(x) \in S] \leq e^{\varepsilon_0} \cdot \Pr[\mathcal{M}(x') \in S].$$

Informally, this definition requires the probability that any adversary can observe a difference when the algorithm operates on any pair of inputs $x$ and $x'$ is bounded by $e^{\varepsilon_0}$.

### 2.2 LDP-SGD

We adopt LDP-SGD (Duchi et al., 2018; Erlingsson et al., 2020) as a randomizing gradient mechanism. The client-side algorithm (Algorithm 1) performs two gradient randomizations in lines 2 and 3. Line 2 is referred to as the **gradient norm projection**, and line 3 is the **random gradient sampling**. We instantiate adversaries based on the following two facts:

**Gradient norm projection**: the smaller the norm of the gradient before randomization, the more likely the sign of the gradient is to be reversed. In addition, the gradient computed in this process forces the norm to be $L$.

**Random gradient sampling**: a gradient close to the gradient before randomization is likely to be generated when privacy parameter $\varepsilon_0$ has a large value. This sampling is illustrated simply in Figure 2. The $\hat{z}$ case is divided into the following cases using Figure 2.

$$\hat{z} = \begin{cases} \text{Sample from the green zone.} & \text{w.p. } \frac{e_0^\varepsilon}{1+e_0^\varepsilon} \\ \text{Sample from the white zone.} & \text{otherwise} \end{cases}$$

### 2.3 PRIVACY AMPLIFICATION VIA SHUFFLING

Shuffle model (Erlingsson et al., 2019; Cheu et al., 2019) anonymizes identities of the randomized responses via secure shuffler and then the anonymization gives us a privacy amplification effect. Let $\varepsilon$ be the amplified privacy after the shuffling. The $\varepsilon$ in the shuffler model is determined by $\varepsilon_0$ set in the local randomizer and the number of client $n$ entering the shuffler. The various interpretations (i.e., privacy amplification theorems) have been proposed. We give the state-of-the-art privacy amplification by shuffling (Feldman et al., 2023) as a theoretical baseline as follows:

$$\varepsilon \le \ln\left(1 + (e^{\varepsilon_0} - 1)\left(\frac{4\sqrt{2\ln 4/\delta}}{(\sqrt{e^{\varepsilon_0}+1})n} + \frac{4}{n}\right)\right). \tag{1}$$

## 3 ANALYSIS OF THE WORST-CASE ATTACK

Here, we analyze the worst-case scenario by considering the characteristics of LDP-SGD. Analyzing worst-case attacks is essential to ensure the mechanism satisfies $\varepsilon$-LDP. Furthermore, at the end of the section, we show the novelties of the LDP-SGD worst-case discovery.

### 3.1 THE WORST-CASE OF LDP-SGD

The worst-case of the Gaussian mechanism (Nasr et al., 2021) involves database contamination and the insertion of a watermark into the gradient. Thus, $g_1$, which has almost all zero components, and $g_2$, which is a watermark $\lambda$ inserted into $g_1$, such as $g_1 = (0, 0, ..., 0)$ and $g_2 = (0, \lambda, ..., 0)$ are the easiest to distinguish. This manipulation is not effective in LDP-SGD due to random gradient sampling. We must ensure that $g_1$ and $g_2$ are unaffected by gradient norm projection and random gradient sampling. In the worst-case, we meet the following proposition:

**Proposition 3.1.** *The worst-case inputs that reach the theoretical upper bound given by $\varepsilon$ are gradients whose $\ell_2$-norm is larger than or equal to $L$ and have a reversed sign.*

We offer the proof in Appendix A.

**Limitation.** Note that the worst-case described in Section 3.1 does not apply to the entire LDP mechanism but only to algorithms with similar properties to LDP-SGD, such as PrivUnit. However, it is different from the existing studies (Nasr et al., 2021; Jagielski et al., 2020; Maddock et al., 2022; Andrew et al., 2023) for the Gaussian mechanisms like DP-SGD under CDP.

### 3.2 JUSTIFICATION OF THE WORST-CASE ATTACK

Figure 3 demonstrates how well the gradient (vector) of the opposite direction can be distinguished after randomization, where $d$ is the dimension of the gradient. $\tilde{g}_1$ means the output of LDP-SGD. Here, the norm of $g_1$ is $L$, the same as the clipping size. Randomization with $\varepsilon = 0.5$ shows that $Cos(\tilde{g}_1, g_1)$ and $Cos(-\tilde{g}_1, g_1)$ overlap and are difficult to distinguish. The reason for the overlap is that when $g_1$ is randomized with $\varepsilon = 0.5$, $Cos(\tilde{g}_1, g_1)$ will be negative with the same probability because the sign is inverted about 38% of the time due to random gradient sampling.

When $\varepsilon$ is set to 4, the cosine similarity is neatly divided around 0, making it easy to distinguish the gradient before randomization. This is because, for $\varepsilon = 4$, the probability of a sign reversal of

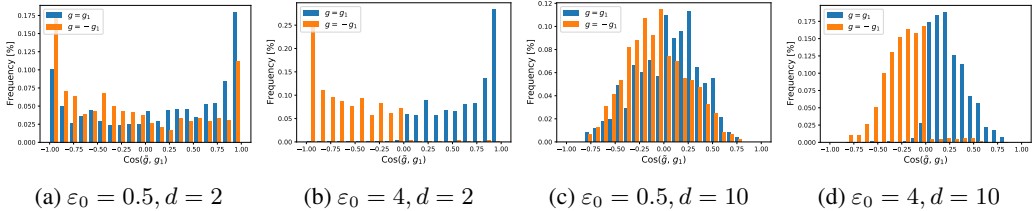

(a) $\varepsilon_0 = 0.5, d = 2$      (b) $\varepsilon_0 = 4, d = 2$      (c) $\varepsilon_0 = 0.5, d = 10$      (d) $\varepsilon_0 = 4, d = 10$

Figure 3: Cosine similarity of gradients before and after randomization. Oppositely oriented gradients tend to show differences in cosine similarity after randomization.

$g_1$ is about 1.8%. The larger the dimension of the vector, the more likely the cosine similarity is to be near 0. However, if the gradient is in the opposite direction, the distribution is clearly divided around 0.

## 4   MEASURING LOWER BOUNDS OF LDP

This section describes the method used to measure the empirical privacy level of FL under LDP. To the best of our knowledge, this is the first attempt to measure the $\varepsilon$ value of LDP empirically.

### 4.1   LOWER BOUNDING $\varepsilon$ OF LDP IN HYPOTHESIS TEST

We follow an analytical approach previously employed to audit CDP (Jagielski et al., 2020; Nasr et al., 2021).

Given an output $y$ of a randomized mechanism $\mathcal{M}$, we consider the following hypothesis testing experiment. We select a null hypothesis as input $x$ and an alternative hypothesis as $x'$:

$$H_0: y \text{ came from input } x$$
$$H_1: y \text{ came from input } x'$$

When selecting rejection region $S$, the false positive rate (FPR), when the null hypothesis was true but rejected, is defined as $\Pr[\mathcal{M}(x) \in S]$. The false negative rate (FNR), when the null hypothesis was false but retained, is defined as $\Pr[\mathcal{M}(x') \in \bar{S}]$, where $\bar{S}$ is a complement to $S$. Mechanism $\mathcal{M}$ that satisfies $\varepsilon$-LDP is equivalent to satisfying the following conditions (Kairouz et al., 2015):

**Theorem 1** (Empirical $\varepsilon_0$-Local Differential Privacy). For any $\varepsilon_0 \in \mathbb{R}^+$, a randomized mechanism $\mathcal{M} : \mathcal{X} \to \mathcal{S}$ is $\varepsilon$-local differentially private if and only if the following conditions are satisfied for any pair of inputs $x, x' \in \mathcal{X}$ and all rejection regions $S \in \mathcal{S}$:

$$\Pr[\mathcal{M}(x) \in S] + e^{\varepsilon_0} \cdot \Pr[\mathcal{M}(x') \in \bar{S}] \geq 1, and \tag{2}$$
$$e^{\varepsilon_0} \cdot \Pr[\mathcal{M}(x) \in S] + \Pr[\mathcal{M}(x') \in \bar{S}] \geq 1.$$

It can be seen that satisfying Equation (2) is equivalent to satisfying $\varepsilon_0$-LDP. Because $1 - \Pr[\mathcal{M}(x) \in S] = \Pr[\mathcal{M}(x) \in \bar{S}]$, which transforms the upper equation, we can obtain $\Pr[\mathcal{M}(x) \in \bar{S}] \leq e^{\varepsilon_0} \cdot \Pr[\mathcal{M}(x') \in \bar{S}]$. Therefore, we can determine the empirical $\varepsilon_0$-local differential privacy as follows:

$$\varepsilon_0^{\text{empirical}} = \max \left( \log \frac{1 - \text{FPR}}{\text{FNR}}, \log \frac{1 - \text{FNR}}{\text{FPR}} \right) \tag{3}$$

For example, in 1000 trials, suppose the FPR when the actual input was $x$ but the distinguisher guessed $x'$ was 0.1, and the FNR when the actual input was $x'$ but the distinguisher guessed $x$ was 0.2. Then, FPR and FNR are substituted into Equation (3) to obtain $\varepsilon_0^{\text{empirical}} \simeq 2.0$.

**Number of trials and measurable $\varepsilon$.** Let us find the number of trials $n$ and the measurable $\varepsilon$ from the Clopper-Pearson bound. We set $\alpha = 0.05$ to get a 95%-confidence bound. The probability of success $p$ is denoted by $\frac{e^{\varepsilon_0}}{1 + e^{\varepsilon_0}}$. Even if the client were to succeed in all 1000 of the trials, the lower confidence limit of $p$ is 0.99632. This is almost equal to the probability of success when $\varepsilon_0$ is 5.6. Similarly, if all 1000000 trials are successful, the Clopper-Pearson bound would imply an $\varepsilon_0$ lower

bound of 12.5. This hypothesis test may be difficult depending on the number of trials and the client machine environment.

## 4.2 Instantiating the LDP adversary in FL

To perform a privacy test based on the hypothesis testing above, we define the following phases:

- **Crafting phase.** The FL client produces two gradients, $g_1$ and $g_2$, with global model $\theta_t$. In this phase, the client honestly randomizes one of the gradients using Algorithm 1 and makes it $\tilde{g}$, which is sent to the distinguishing phase.
- **Distinguishing phase.** The FL client predicts whether the randomized gradient is $g_1$ or $g_2$.

We present this algorithm in Appendix B. We propose the privacy measurement test that repeating crafting and distinguishing a sufficient number of times under specific settings yields $\varepsilon_{\text{empirical}}$ in Equation (3).

## 4.3 Algorithm in distinguishing phase

According to the worst-case attack analysis, the worst-case is achieved by comparing the cosine similarities of the raw and randomized gradients. In this phase, we compare the two raw gradients $\{g_1, g_2\}$ and randomized gradient $\tilde{g}$. Here, we employ the cosine similarity between $\tilde{g}$ and $\{g_1, g_2\}$ for a distinguishing algorithm. The discrimination in the distinguishing phase is defined as follows:

$$guess = \begin{cases} g_1 & cos(\tilde{g}, g_1) \geq cos(\tilde{g}, g_2) \\ g_2 & \text{otherwise} \end{cases}. \tag{4}$$

We employ a case with a higher cosine similarity to the randomized gradient. Calculating the FPR and FNR for this expectation yields $\varepsilon_{\text{empirical}}$.

## 4.4 Algorithms in crafting phase

Although evaluating the effect of every possible combination of assumptions is possible in principle, it is computationally intractable. We must make certain assumptions that will allow computation by the client. We introduce five types of crafting algorithms. Adversaries are classified according to their access levels. The two crafting algorithms assume collusion with the FL server. After generating two candidate inputs for each setting, FL clients choose one gradient from the two with a probability of 50%. The adversary settings proposed in this paper are summarized in Table 2.

**Benign setting.** Under the most realistic settings, suppose that all entities are benign. With this setting, we provide an instantiated adversary with minimal power. In this crafting phase, the client uses the global model $\theta_t$ distributed by the FL server to generate gradients $g_1$ and $g_2$ from inputs $x_1$ and $x_2$, respectively. The input data $x_1$ used for our measurement test is chosen randomly from the dataset. In the case of CIFAR-10, $x_1$ is chosen from 50,000 images. For benign settings, $x_2$ is selected in the same way. Here, we check whether the gradients of two randomly selected inputs are discriminable; for example, $x_1$ is the image of a *cat*, and $x_2$ is the image of a *dog*. Next, $g_1$ or $g_2$ is chosen and randomized in this phase using the client-side algorithm (Algorithm 1).

**Label flip.** The previous setting defined a weak adversary to establish the first baseline lower bound under a realistic setting. The malicious processing of input data includes adding perturbations to the image using FGSM (Goodfellow et al., 2014), Deepfool (Moosavi-Dezfooli et al., 2016), etc. In the case of DP-SGD (Nasr et al., 2021), the poisoned input by Deepfool is the worst-case input for membership inference. The worst-case of LDP-SGD, when only input data poisoning is allowed, can be achieved without using FGSM or Deepfool. To avoid the effects of random gradient sampling, the cosine similarity between $g_1$ and $g_2$ must be negative. The first method that comes to mind to make $g_1$ and $g_2$ face in opposite directions is to flip the sign of $g_1$ to that of $g_2$ (Zhao et al., 2020; Wainakh et al., 2021). Changing the correct labels of the input data is a simple method to approach the worst-case without manipulating the gradients. In this case, a gradient must be generated from a well-trained model. The only difference between $g_1$ and $g_2$ is the data label.

**Proposition 4.1.** *When $\theta_t$ is well pre-trained, changing the input data label can generate a gradient in the opposite direction.*

Table 2: **Adversary settings.** A lower row indicates a stronger adversary.

| Setting | Malicious Access | | $g_1$ | $g_2$ |
| | Crafting phase | FL server | | |
| --- | --- | --- | --- | --- |
| Benign | Benign | Benign | $\nabla \ell(\theta_t; x_1)$ | $\nabla \ell(\theta_t; x_2)$ |
| Label flip | Inputs | Parameters | $\nabla \ell(\theta_t; x_1, label_1)$ | $\nabla \ell(\theta_t; x_1, label_2)$ |
| Gradient flip | Gradients | Benign | $\nabla \ell(\theta_t; x_1)$ | $-g_1$ |
| Collusion | Gradients | Parameters | $\nabla \ell(\tilde{\theta}_t; x_1)$ | $-g_1$ |
| Dummy | Gradients | Benign | $(\lambda, \lambda, \ldots, \lambda)$ | $-g_1$ |

We offer the proof in Appendix C. If $\theta_t$ is not a well-pre-trained model, i.e., the output probability of a positive label is far from 1 in the early stages of FL training, and flipping labels may not work. We must validate $\varepsilon_0^{\text{empirical}}$ for each training stage.

**Gradient flip.** This attack allows FL clients to examine the level of privacy when gradient manipulation is permitted. In the crafting phase, the FL client processes the raw gradient. The simplest method for processing a gradient to increase its discriminability is to flip the gradient sign. As described in Section 3.1, in the random gradient sampling of LDP-SGD, when $\varepsilon_0$ is set to a large value, the sign of the gradient is unlikely to change; therefore, such processing is effective. The client uses the global model, $\theta_t$, distributed by the FL server to generate gradient $g_1$ from input $x_1$. Let $g_2$ be a gradient with a sign that is the opposite of $g_1$.

**Collusion.** Here, we consider distributing a malicious model from the server to the clients. As explained in Section 3.1, in the gradient norm projection of LDP-SGD, a smaller norm for the raw gradient makes it easier for the sign to be flipped. Taking advantage of this property, we consider a setting where the client and server collude, making it challenging to generate a gradient with a small norm. With this setting, the server intentionally creates a global model with massive loss and distributes it to the client. To realize this adversary, we introduce the following procedure. First, the server generates a malicious global model, $\tilde{\theta}_t$, based only on inputs with the same label and distributes it to the client. Second, the client utilizes this malicious model $\tilde{\theta}_t$ to generate gradient $g_1$ from input $x_1$. The label $x_1$ differs from the labels of the inputs used to generate the malicious model. The client flips a gradient in the same manner as gradient flipping.

**Dummy gradient.** In our final and most powerful attack, we assume that the client produces a *dummy* gradient. In other words, FL clients alone can verify the authenticity of LDP-SGD through the worst-case input without collaborating with the server. LDP-SGD includes gradient norm projection, which causes errors owing to random flipping gradients. The simplest way to avoid randomly flipping the sign is to generate a gradient with a large norm, regardless of the FL client's input or model. To craft the dummy gradient, we simply use a constant $\lambda$ to fill all the elements of the gradient. To avoid the gradient norm projection, the norm of $g_1$ must be greater than or equal to the clipping threshold $L$. Therefore, let $\lambda$ be $L/\sqrt{d}$ and $d$ be the dimension of the gradient. As another choice of gradient, we also employ gradient flipping to maximize the difference compared to $g_1$.

## 5 EMPIRICAL PRIVACY IN SHUFFLE MODEL

We also present empirical privacy measures for the shuffle model. The shuffle model of DP has gained significant interest as an intermediate trust model between the standard local and central models (Erlingsson et al., 2019). We argue that from the worst-case of LDP-SGD, the empirical $\varepsilon$ can also be calculated when the gradient is shuffled, as illustrated in Figure 1b. The crucial difference with the local model is that the shuffle model assumes that a mechanism is in place to provide anonymity to each message, i.e., the data collector cannot associate messages with users. This is equivalent to assuming that, in the view of the adversary, these messages have been shuffled by a random permutation unknown to the adversary (Balle et al., 2019).

**Crafting for shuffling.** We define the following neighboring datasets $\mathcal{D}$ and $\mathcal{D}'$. $g_1$ must be greater than or equal to the clipping threshold $L$ to achieve the worst-case of LDP-SGD, and $g_2$ is opposite

to $g_1$. Therefore, the algorithm for generating $g_1$ and $g_2$ is the same as for dummy.

$$\mathcal{D} = \{g_1\}^n; \mathcal{D}' = \{g_1\}^{n-1} + \{g_2\}^1 \tag{5}$$

In Equation (5), let $\mathcal{D}$ denote the case where everyone randomizes the same gradient $g_1$, and let $\mathcal{D}'$ denote the case where only one client randomizes a gradient in the opposite direction $g_2$.

**Distinguishing for shuffling.** Equation (6) distinguishes the input from the shuffler output (the characteristics of the distribution of the gradient), as also shown in Figure 1b. When multi-dimensional vectors are output as in LDP-SGD, the dataset is distinguished by the frequency of positive and negative cosine similarity. When $g_1$ is randomized $n$ times, the expected value of $cos(g, \tilde{g}) > 0$ is $\frac{ne^\varepsilon}{1+e^\varepsilon}$. Therefore, we set $\tau$ in Equation (6) to $\frac{ne^\varepsilon}{1+e^\varepsilon}$.

$$guess = \begin{cases} \mathcal{D} & \sum_{i \in [n]}\{cos(g, \tilde{g}) > 0\} \geq \tau \\ \mathcal{D}' & \text{otherwise} \end{cases} \tag{6}$$

By repeating the trial of predicting the input dataset from the shuffling output, the empirical $\varepsilon$ of the shuffle model can also be derived from Equation (3).

## 6    NUMERICAL OBSERVATIONS

Here, we present the results of an experimental study to observe the numerical results of our LDP tests in FL. Furthermore, we present observations of empirical privacy in the shuffle model. Finally, we discuss the feasibility of a worst-case scenario by FL clients and the possible relaxation of privacy parameter $\varepsilon$. The experimental code is available on the anonymous repository for the blind review [1].

**Experimental setup.** For each adversary instantiation listed in Table 2, we run ten measurements and average the measurement results. Each measurement consists of $K = 1000$ trials. We use LDP-SGD (Erlingsson et al., 2020) to train the models, with clipping norm size $L$=1. The guaranteed privacy levels $\varepsilon$ are set to 0.5, 1.0, 2.0, and 4.0. We observe our measurement test on both image and language models. We utilize LEAF benchmarks (Caldas et al., 2018), which provides benchmark federated datasets for simulating clients with non-IID data and varying local samples. We employ CIFAR-10 (Krizhevsky et al., 2010), FEMNIST (Caldas et al., 2018) and CelebA (Liu et al., 2015) and train a simple CNN as image classification tasks. For our language tasks, we train an LSTM model on non-IID splits of Sent140 (Go et al., 2009). We utilize the FLSim framework [2].

### 6.1    OBSERVATIONS OF EMPIRICAL LDP

Figure 4 summarizes the empirical $\varepsilon$ values for each dataset. Based on tiered access levels, our scenario gives the FL client an assessment of privacy parameter $\varepsilon_0$. Over the four datasets, LDP-SGD was vulnerable to label flipping with a well-pre-trained model and gradient flipping. Adversaries that directly manipulate raw gradients can reach the theoretical upper bounds given by privacy parameter $\varepsilon_0$. In addition, collusion with a server employing the malicious model increases the adversary's capability. Even if only the processing of input data is allowed, the probability of approaching the upper bound increases depending on the classification accuracy of the pre-trained model, i.e., the probability of privacy information leakage increases (Figure 5). If the norm of the dummy gradient is smaller than the clipping size $L$, the gap between the theoretical and measured $\varepsilon_0$ values becomes larger (Figure 6). Details on the execution time of the measurement test are given in Appendix D.

### 6.2    EMPIRICAL PRIVACY AMPLIFICATION BY SHUFFLING

Table 3 shows the theoretical and empirical $\varepsilon$ corresponding to local privacy parameter $\varepsilon_0$ and the number of clients $n$ with $\delta$=$10^{-6}$. We utilize the state-of-the-art privacy amplification Feldman et al. (2023) as the theoretical $\varepsilon$ (Equation (1)). The number of clients required for shuffling is derived from $\varepsilon_0 \leq \ln(\frac{n}{8\ln(2/\delta)} - 1)$ in Feldman et al. (2023). Privacy amplification by shuffling is still in the improvement stage and is expected to become smaller than the value in Equation (1).

---

[1] https://anonymous.4open.science/r/ldp-hypothesis-testing-28FA/
[2] https://github.com/facebookresearch/FLSim

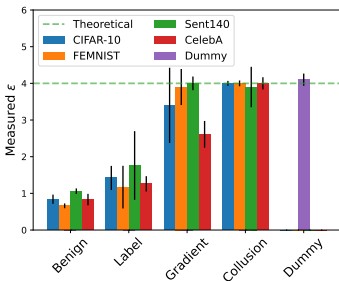

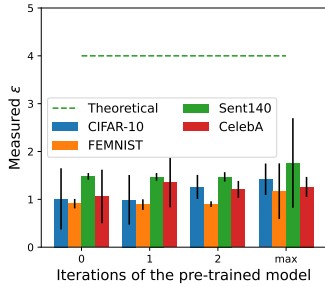

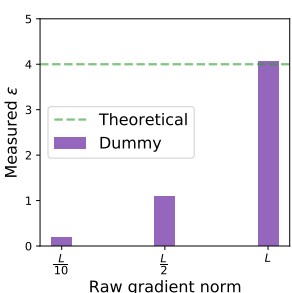

Figure 4: Measured privacy          Figure 5: Iterations and $\varepsilon_{\text{empirical}}$          Figure 6: Effect of gradient norm

Table 3: Theoretical (Feldman et al., 2023) and empirical privacy under shuffle model. Privacy amplification by shuffling may be an improvement over the state-of-the-art.

| $\varepsilon_0$ | Required clients | $n$ | Theoretical $\varepsilon$ | Measured $\varepsilon$ |
|---|---|---|---|---|
| 1 | $n \geq 432$ | 432 | 0.67 | 0.07 |
|   |              | 1000 | 0.48 | 0.06 |
| 2 | $n \geq 974$ | 974 | 0.95 | 0.12 |
|   |              | 1000 | 0.94 | 0.11 |
| 4 | $n \geq 6454$ | 6454 | 1.1 | 0.16 |
|   |              | 10000 | 0.96 | 0.10 |

As shown in Table 3, $\varepsilon_0$=2 used for gradient randomization requires a minimum of 974 clients, and $\varepsilon_0$=4 requires a minimum of 6454 clients for the shuffling.

Empirical privacy amplification is found to be larger than the theoretical value. 974 clients randomize the gradient with $\varepsilon_0 = 2$, and when shuffled, the empirical value is 0.12, despite the theoretical value of $\varepsilon = 0.95$. These results suggest that privacy amplification by shuffling may be an improvement over the state-of-the-art.

### 6.3 DISCUSSION

**Feasibility of a worst-case scenario by FL clients.** An FL client can determine whether the mechanism satisfies $\varepsilon$-LDP by gradient manipulations; however, it might not be allowed to access the gradient directly as a default. An alternative approach is to manipulate the labels, but reaching the theoretical bounds is difficult since it depends on some factors, including the pre-trained model. So, seeking a way to access the gradient is mandatory when we want to assess empirical privacy solidly.

**Relaxations of privacy parameters.** One possible way to prevent worst-case attacks is to prohibit gradient poisoning. As shown in the experiments, the empirical privacy did not reach the given privacy bound without directly manipulating the gradients. This suggests that there may be too much privacy protection in FL under LDP, reducing its utility. By adding restrictions to the FL entities, it may be possible to relax $\varepsilon$.

## 7 CONCLUSION

We introduced a client-side empirical privacy test that measured the lower bounds of LDP. To verify whether the mechanism satisfied $\varepsilon_0$-LDP, we discovered the worst-case of LDP-SGD. In the measurement test of LDP, we demonstrated the empirical privacy levels at various attack surfaces and found that LDP-SGD was vulnerable to label flipping with a well-pre-trained model and gradient flipping. Furthermore, we observed empirical privacy amplification in the shuffle model. To encourage more users to contribute data, our privacy test will contribute to confirm the credibility of the gradient randomized responses in federated learning.

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

## A   PROOF FOR PROPOSITION 3.1

*Proof.* Recall that in LDP-SGD, the direction is reversed, depending on the gradient norm and $\varepsilon_0$. First, the gradient norm projection may flip the gradient sign. From line 2 in Algorithm 1, when a gradient has norm $L$, the probability of preserving the sign is 100%. In other words, a norm smaller than $L$ leads to sign flipping. Hereafter, we assume that $g$ is a gradient with norm $L$ to fix the sign. Next, we recall the random gradient sampling samples from the green zone in Figure 2 with a probability of $\frac{e^{\varepsilon_0}}{1+e^{\varepsilon_0}}$. Unless $g_1$ and $g_2$ are not in the same semicircle (i.e., facing opposite directions), $g_1$ and $g_2$ are difficult to distinguish because of random gradient sampling. We present the effects of sign-flipping on the gradient. Let $g_{\text{flip}}$ be the flipped gradient of $g$, $\tilde{g}$ and $\tilde{g}_{\text{flip}}$ be the outputs of random gradient sampling. In Line 3, there is a $\frac{e^{\varepsilon_0}}{1+e^{\varepsilon_0}}$ probability that $g \cdot \tilde{g} \geq 0$ and $g_{\text{flip}} \cdot \tilde{g}_{\text{flip}} \geq 0$. Let $\tilde{g}_{\text{target}}$ be the gradient randomized to either $g$ or $g_{\text{flip}}$ with a probability of 50%. Let us predict the value of $g_{\text{target}}$. The case can be divided by the cosine similarity between $\tilde{g}_{\text{target}}$ and $g$ as follows:

1. $cos(\tilde{g}_{\text{target}}, g) \geq 0$

   - $g_{\text{target}}$ is $g$ and $g$ rotates within $\pm 90$ degrees. w.p. $\frac{1}{2} \cdot \frac{e^{\varepsilon_0}}{1+e^{\varepsilon_0}}$
   - $g_{\text{target}}$ is $g_{\text{flip}}$ and $g_{\text{flip}}$ rotates over $\pm 90$ degrees. w.p. $\frac{1}{2} \cdot \frac{1}{1+e^{\varepsilon_0}}$

2. $cos(\tilde{g}_{\text{target}}, g) < 0$

   - $g_{\text{target}}$ is $g$ and $g$ rotates over $\pm 90$ degrees. w.p. $\frac{1}{2} \cdot \frac{1}{1+e^{\varepsilon_0}}$
   - $g_{\text{target}}$ is $g_{\text{flip}}$ and $g_{\text{flip}}$ rotates within $\pm 90$ degrees. w.p. $\frac{1}{2} \cdot \frac{e^{\varepsilon_0}}{1+e^{\varepsilon_0}}$

In the distinguishing phase, the FL client predicts $g$ for the original data of $\tilde{g}_{\text{target}}$ when the cosine similarity between $g$ and $\tilde{g}_{\text{target}}$ is positive; then, the prediction is correct with a probability of $\frac{1}{2} \cdot \frac{e^{\varepsilon_0}}{1+e^{\varepsilon_0}}$. Likewise, the client predicts $g_{\text{flip}}$ for the original data of $\tilde{g}_{\text{target}}$ when the cosine similarity between $g$ and $\tilde{g}_{\text{target}}$ is negative; then, the prediction is correct with a probability of $\frac{1}{2} \cdot \frac{e^{\varepsilon_0}}{1+e^{\varepsilon_0}}$. Therefore, the client can distinguish $g$ and $g_{\text{flip}}$ with probability $\frac{e^{\varepsilon_0}}{1+e^{\varepsilon_0}}$. As initially assumed, maximizing the norm and flipping the sign of the gradient are the most effective. $\qquad\square$

## B   ALGORITHM OF OUR MEASUREMENT TEST

---

**Algorithm 2** LDP Test in FL clients

---

**Require:** Privacy parameter: $\varepsilon_0$, #trials: $K$
1: FP, FN, TP, TN $\leftarrow 0$
2: **for** $k \in [K]$ **do**
3:    The FL server sends $\theta_t$ to the client.
4:    **Crafting phase**
5:       $\{g_1, g_2\} \leftarrow \text{Craft}(x_1, x_2, \theta_t)$
6:       Randomly choose $g$ from $\{g_1, g_2\}$
7:       $\tilde{g} \leftarrow \mathcal{A}_{\text{client}}(g)$.
8:       Submit $\tilde{g}$ to the distinguishing phase.
9:    **Distinguishing phase**
10:       guess $\leftarrow \mathcal{D}(\tilde{g}, g_1, g_2)$
11:       **if** $g$ is $g_1$ and guess is $g_2$ **then**
12:          FP += 1
13:       **else if** $g$ is $g_2$ and guess is $g_1$ **then**
14:          FN += 1
15:       **else if** $g$ is $g_2$ and guess is $g_2$ **then**
16:          TP += 1
17:       **else**
18:          TN += 1
19:       **end if**
20: **end for**
21: Compute $\varepsilon_0^{\text{empirical}}$ as (3)

---

## C   Proof for Proposition 4.1

*Proof.* We consider an NN model trained for binary classification with cross-entropy loss ($y = 1$ indicates a positive class, $y = 0$ otherwise):

$$L = -y \log(p) - (1 - y) \log(1 - p)$$

where $p$ is the output probability of an input being labeled positive. More precisely, we write the following:

$$p = \frac{e^{z_+}}{e^{z_+} + e^{z_-}},$$

Where $z_\pm$ is the output (corresponding to the positive and negative classes, respectively) from the penultimate layer before undergoing sigmoid transformation, which we assume to be a linear NN layer: $z_+ = w_+ x + b_+$ and $z_- = w_- x + b_-$, where $w_\pm$ and $b_\pm$ are the weights and biases, respectively, and $x$ is the logit from the previous layer. Assume an input for which the ground-truth label is positive. Then, using the chain rule, the gradient is obtained as follows:

$$\frac{dL}{dW} = \frac{dL}{dp} \cdot \frac{dp}{dz_+} \cdot \frac{dz_+}{dW} = -\frac{1}{p} \cdot p(1-p) \cdot \frac{dz_+}{dW} = (p - 1) \cdot \frac{dz_+}{dW} \tag{7}$$

where $W$ denotes the weight in the NN. If we flip the label (i.e., y = 0), we obtain the gradient

$$\frac{dL}{dW} = \frac{dL}{dp} \cdot \frac{dp}{dz_-} \cdot \frac{dz_-}{dW} = \frac{1}{1 - p} \cdot -p(1-p) \cdot \frac{dz_-}{dW} = -p \cdot \frac{dz_-}{dW} \tag{8}$$

instead. We say that a model is well pre-trained when it is confident that an input with $y = 1$ is positive, i.e., $p \simeq 1$. In other words, given an input with $y = 1$, a well-pre-trained NN would yield a large $z_+$ and small $z_-$. One can then deduce that the weights are trained to satisfy $w_+ \approx -w_-$ such that the above relation holds. It should be noted that $\frac{dz_+}{dW} = \frac{dz_+}{dx}\frac{dx}{dW} = w_+ \frac{dx}{dW}$ and $\frac{dz_-}{dW} = \frac{dz_-}{dx}\frac{dx}{dW} = w_- \frac{dx}{dW}$. When a model is pre-trained, $p \simeq 1$ for $y = 1$, the signs of $p - 1$ in Equation (7) and $-p$ in Equation (8) are the same. We can deduce that the gradient is in the opposite direction when the label is flipped for a well-pre-trained NN. The same argument applies to the inputs with $y = 0$.

□

## D   Running time of measurement test

Figure 7a shows the running time when $\varepsilon_{\text{empirical}}$ is measured from 1000 trials of guessing $g_1$ or $g_2$ on the Laptop (Apple M1 Pro CPU machine with 32 GB memory). Collusion is almost the same process as gradient flip, so it is omitted. Naturally, the dummy provides the fastest measurement since it only puts a constant in $g_1$. The dimension of the dummy setting is the same as FEMNIST, the smallest of the four datasets. The time complexity of our privacy measurement test is proportional to the dimension $d$ of the gradient, i.e., $\mathcal{O}(d)$. The difference in execution time between Datasets is due to differences in the structure of the CNN and LSTM models.

Figure 7b means the estimated execution time of $\varepsilon_0^{\text{empirical}}$ in the dummy setting when the number of trials is between one thousand and one million. As mentioned in Section 4.1, 1000 trials are sufficient to check $\varepsilon_0 = 4$, but if FL clients want to use this hypothesis test to verify that the mechanism guarantees a significant value such as $\varepsilon_0 = 10$, it can be measured by setting the trials at one million. In other words, it takes about 2.2 seconds to check that $\varepsilon_0 = 4$ is guaranteed, while it takes about 37 minutes to verify $\varepsilon_0 = 10$. This experiment was performed on a Laptop, but there may still be issues running it on a smartphone. If the clients verify large $\varepsilon_0$, they may not be able to execute, but a shuffle model can estimate $\varepsilon_0^{\text{empirical}}$ by sharing randomized gradients.

## E   Detailed Measured Empirical Privacy $\varepsilon_0$

Figure 8 shows the empirical $\varepsilon_0$ for each setting in more detail.

**Benign setting.** Simply generating gradients from two different inputs suggests that $g_1$ and $g_2$ would be difficult to use for the worst-case. When we performed this attack on CIFAR-10 and

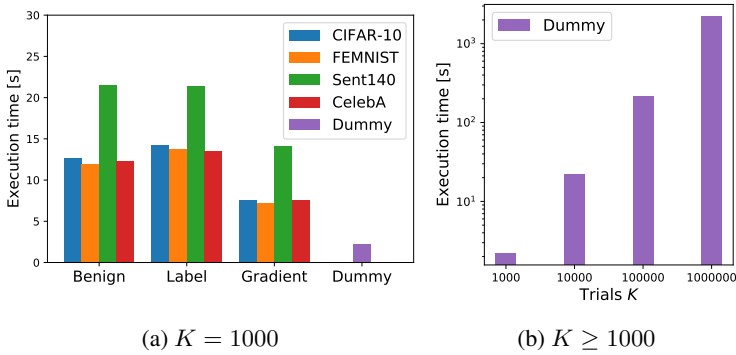

(a) $K = 1000$            (b) $K \geq 1000$

Figure 7: (a) Time required to calculate $\varepsilon_0^{\text{empirical}}$ when trial $K = 1000$. (b) The larger $\varepsilon_0$, the longer the inspection takes.

trained a model with $\varepsilon_0 = 4$, $\varepsilon_{\text{empirical}}$ was 0.85. In other words, the attack accuracy was equivalent to approximately $68\%$. This attack was weak not only when using CIFAR-10 but also when using FEMNIST, Sent140, and CelebA. Therefore, the client requires more ability in the crafting phase to achieve the theoretical upper bound. In the most realistic setting of DP-SGD (Abadi et al., 2016), prior studies (Nasr et al., 2021; Jagielski et al., 2020) have observed the same phenomenon. The distribution of gradients after randomization in this setting is shown in Figure 9a and 9b.

**Label flip.** From Figure 8b, the probability of discrimination increased when manipulating the input label with the well-pre-trained model. The gap between the empirical and theoretical privacy levels in this setting was narrower than that in the benign setting. In other words, label flipping could leverage the poisoning capability to leak more private information. When we performed this attack on Sent140 and trained a model with $\varepsilon_0 = 4$, $\varepsilon_0^{\text{empirical}}$ was 1.76. Figure 5 shows the relationship between the effects of label flipping and the pre-trained models. The horizontal axis represents the number of FL iterations, where the batch size was set to 100. This result indicated that the effect of label flipping became more apparent as the FL training progressed. The well-pre-trained model in this experiment was one that had been trained to the limit in our environment. Depending on the learning method used, it may be closer to the upper bound.

**Gradient flip.** Discriminability improved compared to the benign setting when the client flipped the gradient sign. A comparison of label flip and gradient flip suggests that processing gradients leak more private information than processing input data, especially correct labels. In particular, when $\varepsilon_0 = 4$, $\varepsilon_0^{\text{empirical}}$ had a value of 3.73 on FEMNIST, which was close to the theoretical value. As shown in the worst-case proof, LDP-SGD maintained a direction roughly proportional to the guaranteed $\varepsilon_0$. Thus, $g_1$ and $g_2$, which had opposite signs, could easily be distinguished. However, the empirical value of $\varepsilon$ did not reach its theoretical value for all the datasets because gradients with small norms were occasionally included, causing gradient norm projection.

**Collusion.** Collusion between the client and server narrowed the gap between the theoretical and empirical levels compared with gradient flipping. In this setup, the initial malicious model prevented sign inversion owing to the gradient norm projection, in which gradient flipping was not controlled. The empirical value of $\varepsilon_0$ reached the theoretical value at all setting values, $\varepsilon_0 = 0.5, 1, 2, 4$.

**Dummy gradient.** This setting is a worst-case scenario, and we can ensure that LDP-SGD guarantees a proper $\varepsilon_0$-LDP. From Figure 8e, as with collusion, this setting increases the probability of gradient discrimination. The attack accuracy was approximately $98\%$. Figure 6 shows the effect of the norm on the discrimination probability. Even with a dummy gradient, if the norm is smaller than the clipping size $L$, the gap between the theoretical and measured $\varepsilon_0$ values becomes larger. Therefore, flipping the sign of the gradient with a small norm had little effect. Daring to generate dummy gradients with a large norm is uncommon in FL. However, it was clear that maximizing the norm of the gradient and sign flipping were the most potent attacks.

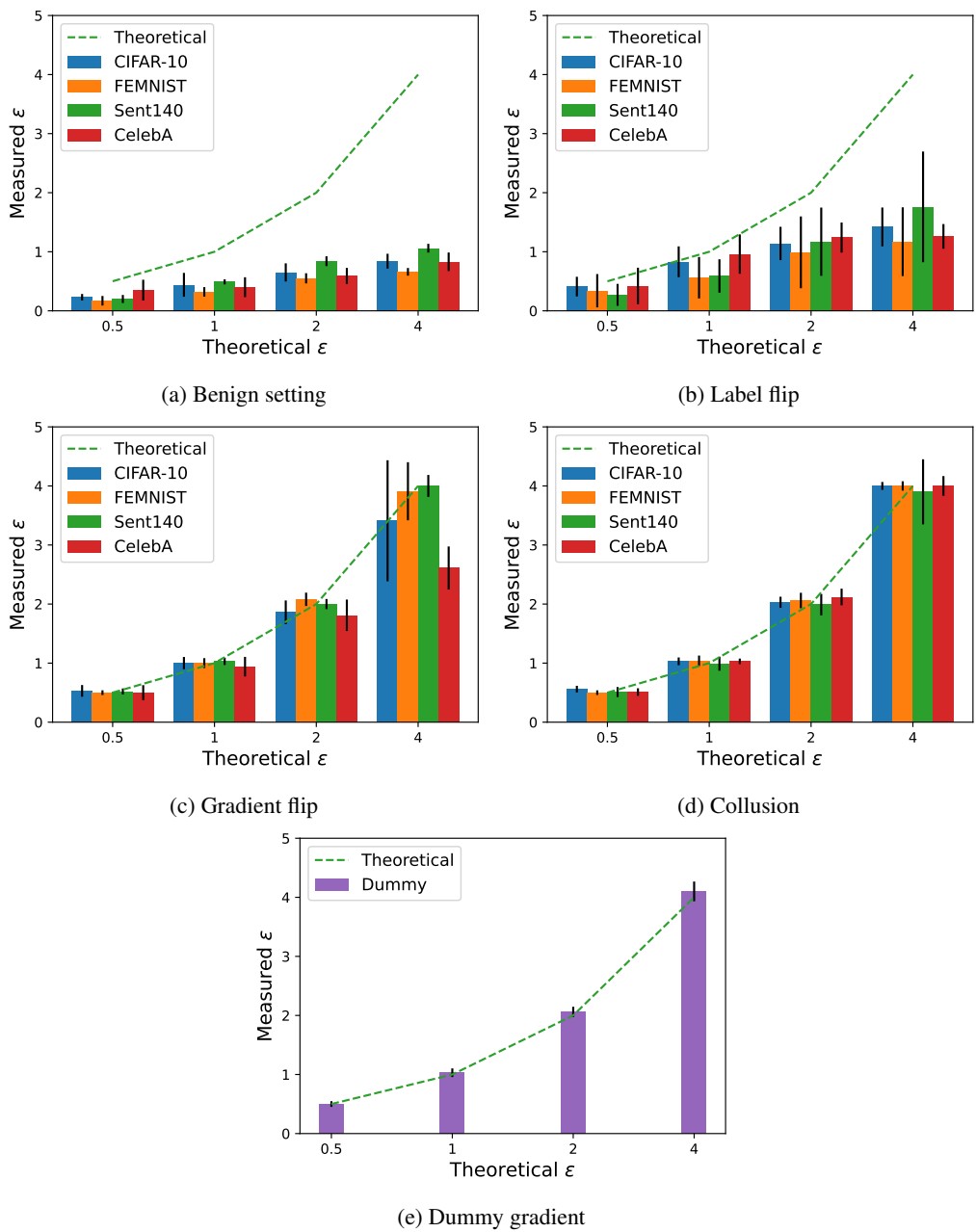

(a) Benign setting

(b) Label flip

(c) Gradient flip

(d) Collusion

(e) Dummy gradient

Figure 8: (a) The client randomizes the gradient without malicious behavior. In practice, the gradient has stronger protection than the assumed privacy level. (b) LDP-SGD is vulnerable to label flipping. (c) LDP-SGD is vulnerable to flipping the gradient sign. Depending on the dataset, $\varepsilon_0^{\text{empirical}}$ almost reached the theoretical value. (d) This setting considers the collusion of the FL client and FL sever to achieve the worst-case. The adversary crafts $g_2$ by flipping $g_1$, just as with gradient flip. (e) The adversary generates a gradient with a large norm, regardless of the image or model possessed by the FL client. $\varepsilon_0^{\text{empirical}}$ reached the theoretical value, confirming that LDP-SGD satisfied $\varepsilon$-LDP.

## F  POISONING EFFECTS

Figure 9 shows how effective the crafting algorithm is in terms of cosine similarity distribution on CIFAR-10. We observe that the stronger the assumed attack, the easier it becomes to distinguish the outputs. We first compare the distributions for $\varepsilon_0 = 0.5$ and $\varepsilon_0 = 4$. It is clear that the larger the epsilon, the more likely there is to be a difference in the post-randomization gradient $\tilde{g}$. We next

show the effect of gradient poisoning. Figure 9a and 9b shows the distribution of cosine similarity in the benign setting, and Figure 9e and 9f show the distribution of cosine similarity in the gradient flip setting. These comparisons show that for $\varepsilon_0 = 4$, the distributions after randomizing $g_1$ and $g_2$ ($-g_1$ for gradient flip) are less likely to mix in gradient flip than in benign setting, making it easier to distinguish.

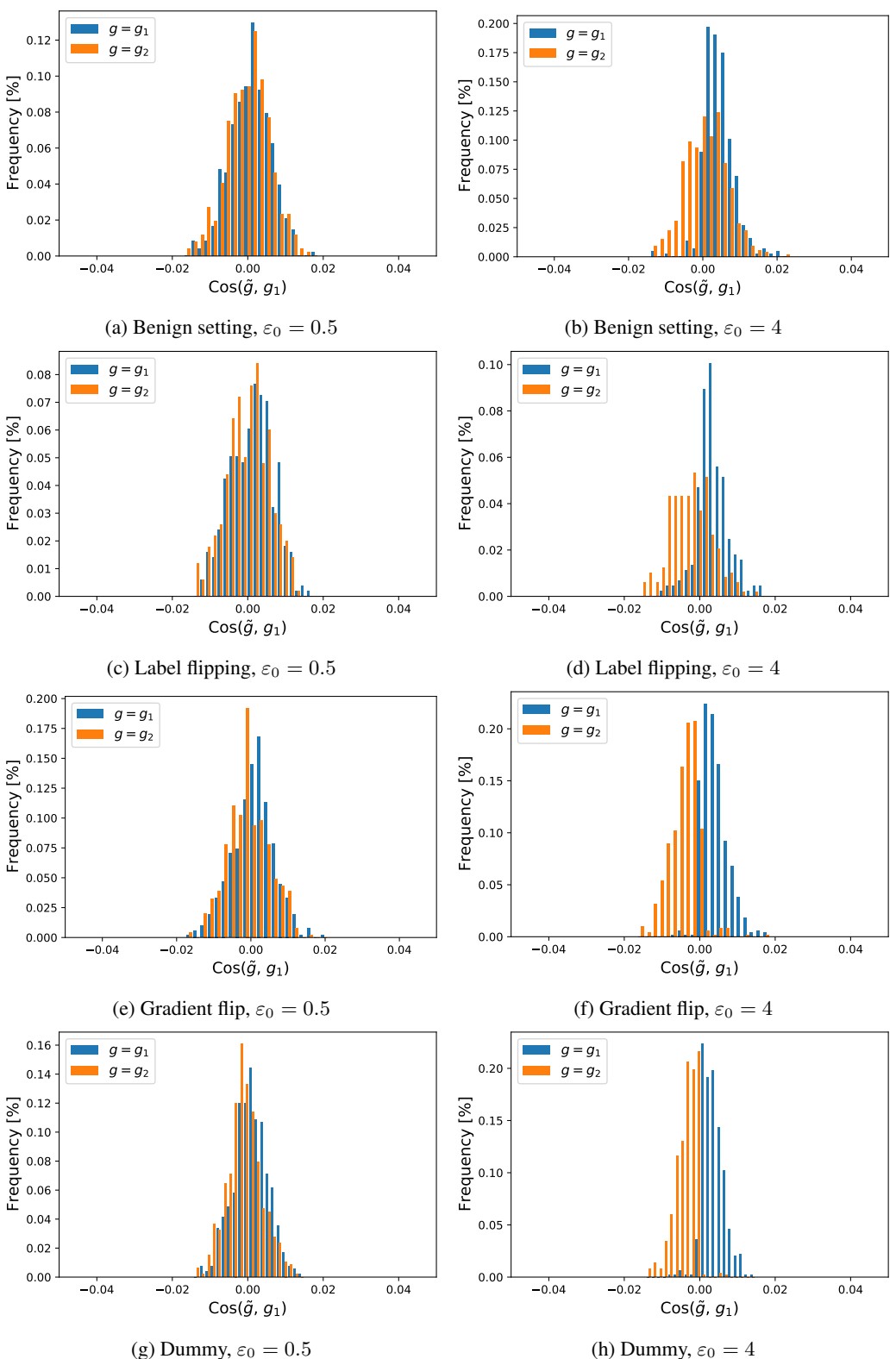

Figure 9: Cosine similarity of gradients before and after randomization. Oppositely oriented gradients tend to show differences in cosine similarity after randomization.

