# OpenReview forum: "Measuring Local and Shuffled Privacy of Gradient Randomized Response"
_ICLR.cc/2024/Conference — ICLR 2024 Conference Withdrawn Submission_

### Official Review · Reviewer_eRua · 2023-10-21

**Soundness:** 2 fair
**Presentation:** 3 good
**Contribution:** 2 fair
**Rating:** 3
**Confidence:** 3

**Summary:**

This paper studies local differential privacy (LDP) guarantees in the federated learning distributed setting. First, the paper empirically considers the privacy loss for the gradient randomized responses including in the well-known LDP-SGD (Duchi et. al. 2018, Erlingsson et. al. 2020), which first computes a clipped gradient and then samples a random unit vector, signed by a function of the clipped gradient as the output. The paper then considers various adversaries in federated learning to produce a worst-case attack that reaches the theoretical limits of LDP-SGD.

The paper first claims that the worst-case inputs that match the theoretical upper bounds of LDP-SGD are achieved when the gradients have norms that match or exceed the clipping threshold $L$, possibly resulting in the incorrect sign. It then shows that for increasing values of the privacy parameter $\varepsilon$, the distributions become easier to distinguish.

The paper then studies a number of attacks, including settings where 1) some labels are flipped, 2) some gradients are flipped, 3) the client and the server collude to flip a gradient, which is compounded by a malicious global model sent from a server, and 4) the client produces a dummy gradient.

**Strengths:**

- Experiments performed over a large number of datasets
- Collusion attack and dummy gradient attack empirically reveal vulnerabilities of LDP-SGD
- The experiments observed privacy amplification through shuffling

**Weaknesses:**

- Both 1) gradients becoming more distinguishable in experiments as the privacy parameter increases and 2) privacy amplification under shuffling matches existing theory and perhaps is not entirely surprising
- Limited conceptual or theoretical novelties

**Questions:**

Were there any characterizations observed for the privacy-convergence or privacy-utility tradeoffs?

---

> ### Author Response · Authors · 2023-11-19
> **Response to Reviewer**
>
> We thank the reviewer for carefully reviewing our paper. We would like to answer the questions you gave.
>
>
>
>
> > Both 1) gradients becoming more distinguishable in experiments as the privacy parameter increases and 2) privacy amplification under shuffling matches existing theory and perhaps is not entirely surprising
>
> In addition to observations of the empirical epsilon, it is also important to discover worst-case scenarios.
>
> The discovery of a worst-case attack allows the client can verify that the randomization algorithm satisfies the epsilon-LDP.
>
> We also suggested the empirical epsilon in the shuffle model (Table 3) that the theoretical value has room for improvement.
>
>
>
>
> > Were there any characterizations observed for the privacy-convergence or privacy-utility tradeoffs?
>
> This may be out of focus as we are not discussing utilities.
>
> If we have misunderstood, please let us know.

---

### Official Review · Reviewer_HQzw · 2023-10-30

**Soundness:** 2 fair
**Presentation:** 3 good
**Contribution:** 3 good
**Rating:** 6
**Confidence:** 4

**Summary:**

The paper looks at empirically measuring differential privacy (DP) level in federated learning (FL) under local DP (LDP) directly, and when the clients jointly communicate all results via a trusted shuffler. The authors focus on the gradient randomized response mechanism as the common LDP mechanism, find a pair of gradients corresponding to the worst-case under the chosen mechanism, and empirically measure how successful membership inference attacks. From the results they convert to empirical ADP guarantees via existing techniques. The paper considers the resutls under 5 different adversaries, including the most powerful adversary allowed by DP.

**Strengths:**

* The topic of empirically measuring DP protection level is important and topical.

* The paper is fairly clear to read.

* While the paper relies on many existing techniques developed for empirical DP estimation, the considered DP mechanism is somewhat different than the existing ones, and the results on shuffle DP seem novel.

**Weaknesses:**

* Comparisons to existing DP bounds in the shuffle model only use the (loose) analytical bound from Feldman et al. 2023.

* In my opinion, the overall story of clients testing by themselves in FL (e.g. abstract) is not convincing or necessary. Instead, this is much more convincing simply as an exploration of empirical privacy in the FL setting, regarless of whether this could be done by the clients bythemselves or by someone else.

* After reading the paper, I still have several questions for the authors (see below for particulars).

**Questions:**

1) First adversary settings (benign, label flip) seem too benign: I would argue that in most cases two random samples from the data will give overpositive results on the privacy level, which is a worst-case guarantee, in many cases possibly even with label flips. Considering this, comments such as in Relaxations of privacy parameters in Sec6.3 seem overconfident. Have you checked what would be the actual worst-case in the data for benign or label-flip? (On a related note, see also question 8)

2) On distinguishing the gradients: do you use cos-similarity  as the similarity metric in all cases, even in the benign setting? Is this still optimal way to do it?

3) Sec 3.1: on the difference in worst-case w.r.t Gaussian mechanism in the centralized setting: despite several statements about the worst case being entirely different compared to the Gaussian mechanism, to me it seems like the given worst case (maximal l2-norm grads pointing to opposite directions) should work as is also for the Gaussian mechanism under replace neighbourhood, and vice versa. Did I misread this?

4) In the empirical shuffling experiments, do you include delta also in the empirical results (it is not included e.g. in Sec 4.1)?

5) Provide some metric of variability for the empirical results, e.g., Fig4, or mention that the variability is small-enough to be ignored.

6) Sec6.2: "These results suggest that privacy amplification by shuffling may be an improvement over the state-of-the-art." What does this mean?

7) Given that we know that the analytical bound for shuffle DP is loose (see e.g. Feldman et al. 2023, Koskela et al 2023), is there some reason not to use numerical accounting when comparing against existing shuffle DP bounds to get at least somewhat tighter bounds?

8) Considering comments like Sec6.3 Relaxations of privacy parameters: how did you choose the norm clipping value for the experiments? One would expect that in the more benign settings choosing a large value leads to small empirical epsilon due to most gradients not hitting the clipping value, i.e., being farther from the worst-case, but this value does not explicitly show up in the results in any way. If so, one could basically choose any empirical epsilon between 0 and the one corresponding to the worst-case as a result of this proposed measurement just by tuning the clipping value. Which of these values would make sense?

#### Minor comments/requests (no reason to comment on this)
i) Please state the neighbourhood definition explicitly in defining DP.

References:
Feldman et al. 2023: Stronger privacy amplification by shuffling...
Koskela et al. 2023: Numerical accounting in the shuffle model of DP

---

> ### Author Response · Authors · 2023-11-19
> **Response to Reviewer Part 1**
>
> We thank the reviewer for carefully reviewing our paper. We would like to answer the questions you gave.
>
> > In my opinion, the overall story of clients testing by themselves in FL (e.g. abstract) is not convincing or necessary. Instead, this is much more convincing simply as an exploration of empirical privacy in the FL setting, regarless of whether this could be done by the clients bythemselves or by someone else.
>
> In collaborative learning like FL, the server often provides a randomized mechanism to all clients to orchestrate the learning process.
> Even when the privacy of the client has been preserved, clients will be concerned about how well the randomized mechanism protects their gradient (or data).
>
> According to a previous survey[xiong2020], some users who did not allow information sharing claimed two reasons: they did not trust the DP techniques and did not trust the app or tech company.
> We believe that to encourage more users to contribute data, it is necessary not only to provide a clear explanation of LDP but also to verify that the randomized mechanism is credible.
>
> [xiong2020] Xiong, Aiping, et al. "Towards effective differential privacy communication for users’ data sharing decision and comprehension." 2020 IEEE Symposium on Security and Privacy (SP). IEEE, 2020.
>
>
>
>
> > First adversary settings (benign, label flip) seem too benign: I would argue that in most cases two random samples from the data will give overpositive results on the privacy level, which is a worst-case guarantee, in many cases possibly even with label flips. Considering this, comments such as in Relaxations of privacy parameters in Sec6.3 seem overconfident. Have you checked what would be the actual worst-case in the data for benign or label-flip? (On a related note, see also question 8)
>
> > Considering comments like Sec6.3 Relaxations of privacy parameters: how did you choose the norm clipping value for the experiments? One would expect that in the more benign settings choosing a large value leads to small empirical epsilon due to most gradients not hitting the clipping value, i.e., being farther from the worst-case, but this value does not explicitly show up in the results in any way. If so, one could basically choose any empirical epsilon between 0 and the one corresponding to the worst-case as a result of this proposed measurement just by tuning the clipping value. Which of these values would make sense?
>
> We thank the reviewer for the helpful comments.
>
> Here, we present additional experimental results on the relationship between the norm clipping value and empirical epsilon.
>
> For FEMNIST dataset and theoretical epsilon = 1,
>
> - Benign setting
>   - clipping size = 1, empirical epsilon = 0.32 (std=0.081)
>   - clipping size = 0.1, empirical epsilon = 0.32 (std=0.053)
> - Label flip
>   - clipping size = 1, empirical epsilon = 0.56 (std=0.351)
>   - clipping size = 0.1, empirical epsilon =0.43 (std=0.135)
>
> From these results, we believe that even if gradient norm projection is avoided in benign setting and label flip, the clipping size is unlikely to affect the empirical epsilon because the gradient is rarely inverted in these scenarios.
>
>
>
>
> > On distinguishing the gradients: do you use cos-similarity as the similarity metric in all cases, even in the benign setting? Is this still optimal way to do it?
>
> As in [Nasr 2021], we also considered a method to compare the increase or decrease in Loss due to the chosen gradient.
>
> However, such a distinguishing algorithm was no more effective than cosine similarity.
>
> This is because, in LDP-SGD (Algorithm 1), the only object that depends on the data is the sign of the inner product in the computation of $\hat{z}$.

---

> > ### Author Response · Authors · 2023-11-19
> > **Response to Reviewer Part 2**
> >
> > > Sec 3.1: on the difference in worst-case w.r.t Gaussian mechanism in the centralized setting: despite several statements about the worst case being entirely different compared to the Gaussian mechanism, to me it seems like the given worst case (maximal l2-norm grads pointing to opposite directions) should work as is also for the Gaussian mechanism under replace neighbourhood, and vice versa. Did I misread this?
> >
> > We describe the worst-case differences between the Gaussian mechanism and LDP-SGD.
> >
> > The worst-case of the Gaussian mechanism involves database contamination and the insertion of a watermark into the gradient.
> >
> > Thus, $g_1$, which has almost all zero components, and $g_2$, which is a watermark inserted into $g_1$, are the easiest to distinguish as shown below:
> >
> > $g_1=(0,0, ..., 0);g_2=(0, \lambda, ..., 0);$
> >
> > Where $\lambda$ is a watermark.
> >
> > This manipulation is not effective in LDP-SGD since, with random gradient sampling, the gradient is sampled from a uniform distribution, and the watermark is removed.
> >
> > Therefore, we must ensure that $g_1$ and $g_2$ are unaffected by gradient norm projection and random gradient sampling.
> >
> >
> >
> >
> > > In the empirical shuffling experiments, do you include delta also in the empirical results (it is not included e.g. in Sec 4.1)?
> >
> > We apologize for the lack of explanation.
> >
> > Given an appropriate $\delta$, we can determine the empirical  ($\epsilon$, $\delta$)−differential privacy as
> >
> > $\epsilon_\text{empirical} = \max(\log\frac{1 − \delta − FPR}{FNR} , \log \frac{1−\delta−FNR}{FPR})$
> >
> > Please also refer to [Nasr 2021].
> >
> > [Nasr 2021] Nasr, Milad, et al. "Adversary instantiation: Lower bounds for differentially private machine learning." 2021 IEEE Symposium on security and privacy (SP). IEEE, 2021.
> >
> >
> >
> >
> > > Provide some metric of variability for the empirical results, e.g., Fig4, or mention that the variability is small-enough to be ignored.
> >
> > We thank the reviewer for the suggestions.
> >
> > Results with error bars were added during the rebuttal period.
> >
> >
> >
> >
> > > Sec6.2: "These results suggest that privacy amplification by shuffling may be an improvement over the state-of-the-art." What does this mean?
> >
> > Let us explain the reasons for the discrepancy between theoretical and empirical values.
> > The current theoretical value of privacy amplification by shuffling is still in the process of development and is uncertain.
> > Therefore, as more research is done to analyze the theoretical value, it is expected to gradually approach the empirical value.
> >
> >
> >
> >
> > > Given that we know that the analytical bound for shuffle DP is loose (see e.g. Feldman et al. 2023, Koskela et al 2023), is there some reason not to use numerical accounting when comparing against existing shuffle DP bounds to get at least somewhat tighter bounds?
> >
> > When considering composition, the results of [Feldman et al. 2023] (using approximate DP and strong composition) are indeed loose. However, since our work does not take composition into account, we do not need to use numerical accounting.
> >
> > Please let us know if anything else is unclear.

---

> > > ### Comment · Reviewer_HQzw · 2023-11-22
> > > **Small clarifications**
> > >
> > > Thanks for the rebuttal, to clarify further:
> > >
> > > i) on the worst-case:
> > > I do not understand your point. For the Gaussian the worst-case under replace neighbourhood assuming sensitivity $2\lambda$ is e.g. $(\lambda, 0, \dots,  0)$ vs $(-\lambda, 0, \dots,  0)$, i.e., grads pointing in opposite directions with norm $L=2\lambda$. You state in the paper that this is exactly the worst-case for LDP-SGD or do you not?
> > >
> > > ii) on the numerical accounting: as far as I can see, the analytical bound is loose even without compositions, see Feldman et al. 2023 (e.g. Sec.6 in v2 on arXiv), Koskela et al. 2023.

---

> > ### Comment · Reviewer_HQzw · 2023-11-22
> > **Further clarification**
> >
> > A small comment on the more benign settings: on a general level, to me this looks something like avg empirical privacy, which I think is not a good proxy for DP, i.e., for the worst-case. The point in the earlier comment was that by setting the clipping bound you can affect how many grads have a high probability of getting flipped, and this is not reflected in any way in the measurement of empirical privacy. If you think about the use case of convincing clients that the mechanism is safe,
> > to me it seems like using your method I could get significantly lower empirical $\epsilon$ in the benign case to convince the clients by setting clipping norm to match the actual worst case for the testing. When you then pick a random sample from the data to check the bound, it is quite likely to not be the worst-case, and since gradients typically are concentrated around 0, I would expect a random sample to have a considerably higher probability of getting flipped than the worst-case. Is this reasoning wrong?
> >
> > Please also clarify in the paper explicitly when your proposed method is optimal and when it is not.

---

### Official Review · Reviewer_MdJg · 2023-10-31

**Soundness:** 3 good
**Presentation:** 2 fair
**Contribution:** 2 fair
**Rating:** 5
**Confidence:** 3

**Summary:**

This work extends the line of privacy auditing research to LDP. They analyze the worst-case gradient pair of LDP-SGD mechanism, and use the worst-case pair to design a simple distinguishing game for measuring the lower bound of epsilon through the classic Clopper-Pearson bound. The paper discusses different capability of the adversary. The paper also extends the attack method to the shuffle model.

**Strengths:**

The paper has a clear roadmap and is very easy to follow.

**Weaknesses:**

1. I am not too sure about the motivation of privacy auditing for LDP. The point of LDP is that the clients do not trust the central server and want to do the randomization on their own. I think the authors' intention was to verify the privacy guarantee of the local randomizer the client uses, but why can't the clients just run the local randomization program they trust (e.g., the one implemented by themselves)? I think the authors should be very clear about the motivation and assumptions for the scenario they consider here.

2. The technical contribution of this work is relatively low. My feeling is that the only notable contribution in this work is the discovery of the worst-case gradient pair for the LDP-SGD algorithm (and the result for that is also not too surprising). Happy to be corrected on this point.

3. The paper has quite a few places that lack mathematical rigor. For example, I didn't find where the author defines $\tilde{g}_1$ and $\tilde{g}_2$ in Section 3.2. The assumption for Proposition 4.1 should be clearly stated (the loss is binary cross-entropy). Also, it seems equation 6 is missing something (I guess it's indicator function)?

4. I am not entirely sure about what is the message of Section 3.2 is trying to convey. I feel like it's quite obvious that when $\epsilon$ is small, it's hard to distinguish, and when $\epsilon$ is large, it's easier to distinguish. What sounds interesting is the difference when using different $d$, but I don't find a discussion for it in the paper.

**Questions:**

Does the dimension $d$ impact the attack performance (if fix the number of trials)?

---

> ### Author Response · Authors · 2023-11-16
> **Response to Reviewer**
>
> We thank the reviewer for carefully reviewing our paper. We would like to answer the questions you gave.
>
> > I am not too sure about the motivation of privacy auditing for LDP. The point of LDP is that the clients do not trust the central server and want to do the randomization on their own. I think the authors' intention was to verify the privacy guarantee of the local randomizer the client uses, but why can't the clients just run the local randomization program they trust (e.g., the one implemented by themselves)? I think the authors should be very clear about the motivation and assumptions for the scenario they consider here.
>
> In collaborative learning like FL, the server often provides a randomized mechanism to all clients to orchestrate the learning process.
> Even when the privacy of the client has been preserved, clients will be concerned about how well the randomized mechanism protects their gradient (or data).
>
> According to a previous survey[xiong2020], some users who did not allow information sharing claimed two reasons: they did not trust the DP techniques and did not trust the app or tech company.
> We believe that to encourage more users to contribute data, it is necessary not only to provide a clear explanation of LDP but also to verify that the randomized mechanism is credible.
>
>
>
>
> > The technical contribution of this work is relatively low. My feeling is that the only notable contribution in this work is the discovery of the worst-case gradient pair for the LDP-SGD algorithm (and the result for that is also not too surprising). Happy to be corrected on this point.
>
> We also introduce a way to measure empirical privacy amplification by shuffling.
>
> The empirical epsilon in the shuffle model we present in Table 3 suggests that the theoretical value has room for improvement.
>
>
>
>
> > The paper has quite a few places that lack mathematical rigor. For example, I didn't find where the author defines \tilde{g_1} and \tilde{g_2} in Section 3.2. The assumption for Proposition 4.1 should be clearly stated (the loss is binary cross-entropy). Also, it seems equation 6 is missing something (I guess it's indicator function)?
>
> We apologize for the lack of explanation.
>
>  means the output of the LDP-SGD.
>
> Due to space limitations, the assumption for Proposition 4.1 is shown in Appendix C.
>
> Similar to Equation 4, Equation 6 is the distinguishing algorithm.
>
> Equation 6 distinguishes the input from the shuffler output (the characteristics of the distribution of the gradient), as also shown in Figure 1(b).
>
> Explanations for these additions were added during the rebuttal period.
>
>
>
>
> > I am not entirely sure about what is the message of Section 3.2 is trying to convey. I feel like it's quite obvious that when epsilon is small, it's hard to distinguish, and when epsilon is large, it's easier to distinguish. What sounds interesting is the difference when using different epsilon, but I don't find a discussion for it in the paper.
>
> > Does the dimension d impact the attack performance (if fix the number of trials)?
>
> We apologize for the lack of explanation.
>
> First, there is the property that higher dimensional vectors have lower cosine similarity.
>
> However, if the gradient is in the opposite direction, the distribution is clearly divided around 0.
>
> This suggests that determining a non-zero threshold and predicting the original gradient from the cosine similarity may not discriminate well depending on the dimension.
>
>
>
>
> [xiong2020] Xiong, Aiping, et al. "Towards effective differential privacy communication for users’ data sharing decision and comprehension." 2020 IEEE Symposium on Security and Privacy (SP). IEEE, 2020.
>
>
>
>
> Please let us know if anything else is unclear.

---

### Official Review · Reviewer_MFPE · 2023-11-01

**Soundness:** 1 poor
**Presentation:** 3 good
**Contribution:** 1 poor
**Rating:** 3
**Confidence:** 2

**Summary:**

This paper studies schemes for auditing the empirical privacy parameters of the LDP mechanism (Specifically PrivUnit) and the shuffled model. The idea is to estimate the false positive rate (FPR) and true positive rate (TPR) by running the PrivUnit mechanism multiple times with half of the runs on gradient $g_1$ and the other half of runs on gradient $g_2=-g_1$. Then, the empirical estimate of $\varepsilon_0$ is obtained from the estimated FPR and TPR. This is exactly the standard black-box algorithm in estimating the empirical $\varepsilon$ in the central DP (except the worst case neighboring datasets).

**Strengths:**

The LDP and the shuffled model can be seen as a special case of the central DP mechanisms. In other words, the $\varepsilon_0$-LDP mechanism is also $\varepsilon_0$-DP in the central DP model ( similarly for the shuffled model). Hence, I don't understand the main difference between the scheme proposed in this paper and the schemes in the literature for CDP.

**Weaknesses:**

I disagree with the authors in the introduction that the previous studies estimate the empirical privacy of the Gaussian mechanism. There are some studies that consider the general $\varepsilon$-DP mechanism, e.g., Algorithm 2 in [Jagielski 2020] is generic and isn't dedicated to the Gaussian mechanism. Also [Steinke 2023] proposed a scheme for empirically estimating $\varepsilon$ for a generic DP mechanism in $\mathcal{O}(1)$ training run.

It is not clear to me the novelty of this work.  The paper combines ideas from the existing work in the literature. The techniques used for CDP can be applied to empirically estimate $\varepsilon_0$ in LDP and $\varepsilon$ in the shuffled model. The only difference is in constructing the worst-case neighboring data points in the LDP which is straightforward for the considered PrivUnit mechanism [Duchi et. al. 2018].

The paper focuses mainly on empirically estimating $\varepsilon_0$ of a special version of the PrivUnit mechanism [Duchi et. al. 2018 ] which is order optimal only in the high privacy regime $\varepsilon_0\leq 1$. It is better to focus on the PrivUnit2  [Bhowmick 2019] which is optimal for all privacy regimes. What about a generic $\epsilon_0$-LDP mechanism? Are there any ideas on how to handle this case?







Matthew Jagielski, Jonathan Ullman, and Alina Oprea. Auditing differentially private machine learning: How private is private sgd? Advances in NeurIPS 2020

Steinke, Thomas, Milad Nasr, and Matthew Jagielski. "Privacy Auditing with One (1) Training Run." arXiv preprint arXiv:2305.08846 (2023).

**Questions:**

Please, check my comments in the weaknesses section.

---

> ### Author Response · Authors · 2023-11-16
> **Response to Reviewer**
>
> We thank the reviewer for the helpful comments. Please find our replies to the comments below.
>
> > There are some studies that consider the general \epsilon-DP mechanism, e.g., Algorithm 2 in [Jagielski 2020] is generic and isn't dedicated to the Gaussian mechanism. Also [Steinke 2023] proposed a scheme for empirically estimating epsilon for a generic DP mechanism in O(1) training run.
>
> Indeed, the methods of [Jagielski 2020] and [Steinke 2023] are applicable to any mechanism.
>
> However, in those papers, only the empirical epsilon of the Gaussian mechanism is observed, and the worst-case values for the Gaussian mechanism are taken into account.
>
> Since the worst-case of LDP-SGD is different from that of the Gaussian mechanism, the empirical epsilon presented by [Jagielski 2020] and [Steinke 2023] does not yield similar results for any mechanism.
>
>
>
> We claim that our contribution is that we devised the adversary for mechanisms like LDP-SGD and PrivUnit.
>
> However, observing empirical epsilon alone is not enough; it is also important to discover the worst-case scenario.
>
> The discovery of a worst-case attack allows the client can verify that the randomization algorithm satisfies the epsilon-LDP.
>
> According to a previous survey[xiong2020], some users who did not allow information sharing claimed two reasons: they did not trust the DP techniques and did not trust the app or tech company.
> We believe that to encourage more users to contribute data, it is necessary not only to provide a clear explanation of LDP but also to verify that the randomized mechanism is credible.
>
> We also introduced a way to measure empirical privacy amplification by shuffling.
>
> The empirical epsilon in the shuffle model we present in Table 3 suggests that the theoretical value has room for improvement.
>
>
>
> [xiong2020] Xiong, Aiping, et al. "Towards effective differential privacy communication for users’ data sharing decision and comprehension." 2020 IEEE Symposium on Security and Privacy (SP). IEEE, 2020.
>
>
> Please let us know if anything else is unclear.

---

### Official Review · Reviewer_Hrxc · 2023-11-10

**Soundness:** 2 fair
**Presentation:** 3 good
**Contribution:** 2 fair
**Rating:** 5
**Confidence:** 3

**Summary:**

This paper proposes an approach enabling a user to locally compute a lower bound on the privacy provided through an approach such as gradient randomized response, which is a mechanism ensuring local differential privacy in stochastic gradient descent. More precisely, the setting considered is that of federated learning and the objective is to be able to audit the privacy guarantees, thus obtaining a lower bound on epsilon through the use of privacy attacks conducted on the client-side. The approach proposed can also be adapted to the privacy via shuffling technique.

**Strengths:**

The authors have done a good review of the related work on privacy auditing and clearly position their work with respect to the state-of-the-art. They have also clearly explained how the LDP-SGD algorithm works and conduct a thorough analysis of the worst-case attack. I have particularly appreciate Figure 3, which clearly illustrates the impact of the privacy parameter on the distinguishability of gradients.

One of the main contribution of the paper is the proposition of five crafting algorithms for the auditing phase. These approaches have been tested with a wide range of datasets. The results obtained clearly demonstrate that some of these approaches provide non-trivial lower bound when epsilon is large.

**Weaknesses:**

Although Table 2 aims at classifying the different attacks proposed in terms of adversary power, this issue should be discussed more in the paper. In particular, it is not clear for me about all the assumptions that are needed in practice for these attacks to be implemented in real-life. It would be great of the authors could expand a bit more on these aspects. The impact on the utility of the model of these different approaches should also be discussed.

With respect to the experiments conducted, more details are needed to be able to understand them. For instance, in Figure 4 for the collusion attack it seems that the measured privacy is above the theoretical one for FEMNIST and CelebA, which seems strange. In addition, in Table 3, the measured epsilons seem to be too low to provide a meaningful theoretical guarantee.

A few typos :
-« our proposed privacy test has a novelty to discuss » -> « one of the novelty of our proposed privacy test is that it discusses »
-« We utlize the state-of-the-art privacy » -> « We utilize the state-of-the-art privacy »
-« Recuired clients » -> « Required clients »
-« may still be issues running on a smartphone » -> « may still be issues running it on a smartphone »
-« poising effects » -> « poisoning effects »

**Questions:**

Please see the main comments in the weaknesses section.
One additional question : How is the reference Evfimievski 2003 also related to local differential privacy as cited in the introduction?

---

> ### Author Response · Authors · 2023-11-16
> **Response to Reviewer**
>
> Thank you for your valuable review.
>
> We address some of your specific concerns below.
>
> > Although Table 2 aims at classifying the different attacks proposed in terms of adversary power, this issue should be discussed more in the paper. In particular, it is not clear for me about all the assumptions that are needed in practice for these attacks to be implemented in real-life. It would be great of the authors could expand a bit more on these aspects. The impact on the utility of the model of these different approaches should also be discussed.
>
>
>
>
> We explain what each attack means in real life.
>
> First of all, clients may be restricted from accessing memory even on their own devices.
>
> Gradient flip/collusion/dummy assumes that memory access is allowed and that the client can modify the gradient.
> On the other hand, the benign setting is for the case where no data access is allowed, and label flip is for the case where only the input data to the gradient is accessible.
>
>
>
>
> > With respect to the experiments conducted, more details are needed to be able to understand them. For instance, in Figure 4 for the collusion attack it seems that the measured privacy is above the theoretical one for FEMNIST and CelebA, which seems strange.
>
> With epsilon=4, the error is 0.84% for a confidence level of 95% and 1000 trials, so there are cases where the empirical value is slightly higher than the theoretical value.
>
> The more attempts the client makes, the smaller this error will be.
>
>
>
>
> > In addition, in Table 3, the measured epsilons seem to be too low to provide a meaningful theoretical guarantee.
>
> Let us explain the reasons for the discrepancy between theoretical and empirical values.
> The current theoretical value of privacy amplification by shuffling is still in the process of development and is uncertain.
> Therefore, as more research is done to analyze the theoretical value, it is expected to gradually approach the empirical value.
>
>
> We apologize for the typo.
>
> Please let us know if anything else is unclear.

---

> > ### Comment · Reviewer_Hrxc · 2023-11-22
> >
> > I thank the authors for their answer that have clarified some issues I had. However, based on the many other important points raised by other reviewers, I am not updating my score.